# Effect of Upper Airway Stimulation in Patients with Obstructive Sleep Apnea (EFFECT): A Randomized Controlled Crossover Trial

**DOI:** 10.3390/jcm10132880

**Published:** 2021-06-29

**Authors:** Clemens Heiser, Armin Steffen, Benedikt Hofauer, Reena Mehra, Patrick J. Strollo, Olivier M. Vanderveken, Joachim T. Maurer

**Affiliations:** 1Department of Otorhinolaryngology/Head and Neck Surgery, Klinikum Rechts der Isar, Technical University of Munich, 81675 München, Germany; b.hofauer@tum.de; 2Department of Otorhinolaryngology/Head and Neck Surgery, University of Luebeck, 23562 Luebeck, Germany; Armin.Steffen@uksh.de; 3Department of Sleep Medicine, Cleveland Clinic Foundation, Cleveland, OH 44195, USA; MEHRAR@ccf.org; 4Department of Pulmonary, Allergy and Critical Care Medicine, University of Pittsburgh, Pittsburgh, PA 15261, USA; strollopj@upmc.edu; 5Multidisciplinary Sleep Disorders Centre, Antwerp University Hospital, 2650 Edegem, Antwerp, Belgium; Olivier@ovanderveken.be; 6Department of ORL-HNS, Division of Sleep Medicine, University Medical Centre Mannheim, University Heidelberg, 68167 Mannheim, Germany; joachim.maurer@umm.de; 7Department of Information Technology, University of Applied Sciences, 68163 Mannheim, Germany

**Keywords:** hypoglossal nerve stimulation, obstructive sleep apnea, upper airway stimulation, surgical treatments, randomized trial

## Abstract

Background: Several single-arm prospective studies have demonstrated the safety and effectiveness of upper airway stimulation (UAS) for obstructive sleep apnea. There is limited evidence from randomized, controlled trials of the therapy benefit in terms of OSA burden and its symptoms. Methods: We conducted a multicenter, double-blinded, randomized, sham-controlled, crossover trial to examine the effect of therapeutic stimulation (*Stim*) versus sham stimulation (*Sham*) on the apnea-hypopnea index (AHI) and the Epworth Sleepiness Scale (ESS). We also examined the Functional Outcomes of Sleep Questionnaire (FOSQ) on sleep architecture. We analyzed crossover outcome measures after two weeks using repeated measures models controlling for treatment order. Results: The study randomized 89 participants 1:1 to *Stim* (45) versus *Sham* (44). After one week, the AHI response rate was 76.7% with *Stim* and 29.5% with *Sham*, a difference of 47.2% (95% CI: 24.4 to 64.9, *p* < 0.001) between the two groups. Similarly, ESS was 7.5 ± 4.9 with *Stim* and 12.0 ± 4.3 with *Sham*, with a significant difference of 4.6 (95% CI: 3.1 to 6.1) between the two groups. The crossover phase showed no carryover effect. Among 86 participants who completed both phases, the treatment difference between *Stim* vs. *Sham* for AHI was −15.5 (95% CI −18.3 to −12.8), for ESS it was −3.3 (95% CI −4.4 to −2.2), and for FOSQ it was 2.1 (95% CI 1.4 to 2.8). UAS effectively treated both REM and NREM sleep disordered breathing. Conclusions: In comparison with sham stimulation, therapeutic UAS reduced OSA severity, sleepiness symptoms, and improved quality of life among participants with moderate-to-severe OSA.

## 1. Introduction

Obstructive sleep apnea (OSA) is a common and under-recognized disease in western industrialized countries. In the United States, the estimated prevalence of moderate-to-severe OSA in those 30–70 years old is 6% in women and 13% in men [1]. In the HypnoLaus study from Switzerland, Heinzer et al. reported mild OSA in nearly 40% of men under 60 years of age [2]. The Wisconsin Sleep Cohort Study, established over two decades ago, demonstrated a relationship between OSA and obesity, thus, as obesity increases globally, the incidence of OSA is expected to increase as well [3]. The standard treatment for OSA is continuous positive airway pressure (CPAP), which is effective but fraught with challenges of maintaining adherence [4]. Other treatment options include mandibular advancement devices (MAD), weight loss, behavior modifications, and surgical options [4,5].

Breathing-cycle-synchronized selective upper airway stimulation (UAS) has evolved as a viable treatment for OSA patients intolerant of CPAP [6,7]. UAS targets the loss of upper airway muscle tone during sleep, mainly in the genioglossus muscle [6,8,9]. Several multicenter prospective clinical trials have demonstrated the effectiveness of UAS in participants with moderate-to-severe OSA [10,11,12,13,14,15,16,17]. The initial multicenter Stimulation Therapy for Apnea Reduction (STAR) trial in 2014 demonstrated a decrease in the apnea–hypopnea index (AHI) from 32.0/h at baseline to 15.3/h after 12 months (*p* < 0.001) [10]. Additional data from this cohort showed a sustained AHI reduction after 5 years [15]. Several additional multicenter international prospective studies subsequent to the STAR trial have reported the consistent effectiveness of UAS as well as a favorable safety profile and patient receptivity [16,17,18,19].

The STAR trial included a randomized withdrawal study arm performed after 12 months. By protocol, the first 46 successfully treated participants were randomized to either therapy maintenance (therapy remained active) or therapy withdrawal. After one week, participants were then reassessed with an in-lab polysomnography (PSG) [20]. The AHI in the therapy withdrawal group increased to the levels observed before surgery, while the AHI in the therapy maintenance group remained stable. The trial’s main limitation was only including participants who were responders to therapy. To address this limitation, we designed a randomized control trial to prospectively enroll UAS recipients regardless of their therapy response. The primary endpoints were the improvement in sleep disordered breathing measured by the AHI and self-reported daytime sleepiness assessed by the Epworth Sleepiness Scale (ESS). Secondary endpoints included the change in sleep-related quality of life using the Functional Outcomes of Sleep Questionnaire (FOSQ) and the Clinical Global Impression of Improvement (CGI-I) by the treating investigators, as well as the differential impact of UAS on sleep-disordered breathing in NREM versus REM sleep.

## 2. Materials and Methods

*Trial design and participants* The effect of Upper Airway Stimulation in patients with OSA (EFFECT) trial was a multicenter, double-blinded, randomized, sham-controlled, crossover study. All participants were recruited from three clinical centers in Germany (Mannheim, Munich, and Luebeck). Klinikum rechts der Isar, Technical University of Munich, coordinated and managed the trial. The relevant regulatory authorities and ethics committees at each participating site approved the protocol. The crossover study design assessed the treatment effect of UAS at two different time points with two therapy settings. The study flow chart is depicted in Figure 1. The study was registered at clinicaltrials.gov (NCT03760328).

Participants received implantation of UAS (Inspire Medical Systems, Golden Valley, MN, USA) at least six months prior to enrollment. The main inclusion criteria for UAS were moderate-to-severe OSA (AHI ≥ 15), CPAP intolerance, and the absence of complete concentric retropalatal collapse during drug-induced sleep endoscopy. All recipients of UAS between 2014 to 2019 were eligible for recruitment and were *consecutively* recruited regardless of whether they were responders or non-responders to therapy according to the Sher criteria [21]. All participants gave written informed consent.

*Randomization and masking* Upon completion of the baseline PSG with therapy ON, each participant was randomized 1:1 to one of two groups: therapeutic stimulation (*Stim*) or sham stimulation (*Sham*) using a centralized, computer-generated, password-protected system. The UAS devices implanted in the participants were then programmed to the setting assigned to their respective groups, i.e., *Stim* (continued therapeutic stimulation, average amplitude 1.6 V ± 0.7) and *Sham* (stimulation voltage set at 0.1 V as a subtherapeutic stimulation level and a deception for the patient). 

The sleep technician at each center randomized the sequence of the intervention participants were exposed to and programmed their devices without the knowledge of the participant or physician investigator, who remained blinded to the randomization status. The PSGs were analyzed and scored by another sleep technician, who was blinded during all procedures. 

*Procedures* The crossover design of the EFFECT study included three separate visits at intervals of one week. All participants received therapeutic stimulation during the first visit (baseline visit). After receiving randomized assignment, the *Stim–Sham* group received therapeutic stimulation while the *Sham–Stim* group received sham stimulation for one week. During the second week, the *Stim–Sham* group received sham stimulation while the *Sham–Stim* group received therapeutic stimulation. At each of the three study visits, participants underwent PSG. AHI and oxygen desaturation index (ODI) were scored using standard 2017 scoring criteria of the AASM, with hypopnea scored according to 30% airflow reduction and 4% oxygen desaturation [22]. At each visit, participants completed a standard medical history and demographic survey that documented body mass index (BMI), sex, blood pressure, race, current medications, alcohol use, a functional tongue exam, snoring history, and a Clinical Global Impression-Improvement (CGI-I) assessment. Participants also completed two questionnaires, ESS and FOSQ, and a Patient Satisfaction Survey (PSS) [23,24]. 

*Study Outcome Measures* The study had two co-primary endpoints. The first was the proportion of AHI responders between parallel randomized groups at the 1-week visit. AHI responder was defined as AHI ≤ 15/h. The second co-primary endpoint was the self-reported sleepiness measure using the ESS questionnaire at the 1-week visit. If the primary efficacy endpoints were met, the endpoints were then analyzed according to the crossover design. For additional outcome measures, participants completed the FOSQ, a quality-of-life questionnaire designed specifically to evaluate the impact of excessive sleepiness on activities of daily living, at every visit. Clinicians assessed participants using the CGI-I scale to measure the severity of participants’ overall improvement with the intervention. The 7-point CGI-I scale requires the clinician to assess how much a participant’s illness has improved or worsened relative to a baseline state at the beginning of the intervention. Finally, sleep data on each participant was collected at every visit using an in-lab PSG. The recorded data was converted and scored for analysis by a blinded independent sleep technician at each site. 

*Statistical analysis* Sample size was conservatively estimated for the parallel group comparisons of the 1-week endpoints. For the primary endpoints, a one-sided chi-square test of superiority with a superiority margin of 10% was used with the following assumptions: one-sided Type I error of 0.25, power of ≥ 80%, expected response rates of 70% with continued therapeutic stimulation and 30% with sham stimulation with a superiority margin ∆ of 10%. Under these assumptions, the required minimum sample size was 84 participants (42 per group). The significance level of 0.025 was based on a two-sided Type I error rate of 5%. A one-sided test was performed because we were testing the expected response rate with a superiority margin. For the co-primary endpoint, self-reported sleepiness, the test was a one-sided test of null improvement of 2 points using a t-test under the following assumptions: overall one-sided Type I error of 0.025 and power ≥80%. Under these assumptions, the required minimum sample size was 24 participants (12 per group). The required minimum sample size was based on the observed improvement in the STAR trial, assuming the same means and standard deviations of ESS of 5.6 ± 3.9 with therapy ON and 10.0 ± 6.0 with therapy OFF [10,20]. 

We analyzed crossover outcome measures after 2 weeks using repeated measures models controlling for treatment order. To compare the PSG characteristics, ESS and FOSQ between Stim versus Sham, we used a random effects model including the baseline value as a covariate and controlling for testing order. P-values shown reflect the test of difference between Stim and Sham in changes from baseline.

## 3. Results

A total of 89 participants were assessed for eligibility and randomized between December 2018 and November 2019 (Figure 1). After the baseline visit with therapy, 45 participants were randomized to the Stim–Sham group and 44 participants to the Sham–Stim group. Three participants did not complete the study: two participants from the Stim–Sham group were lost to follow-up prior to the 1-week visit. One participant from the Sham–Stim group exited the study prior to the 2-week visit due to a stroke that was deemed unrelated to UAS. Baseline characteristics of the study cohort revealed that the two groups were well-balanced in baseline characteristics (Table 1). The participants were middle-aged and mildly obese with moderate-to-severe OSA. The average UAS use in the Stim–Sham group was 33.9 ± 22.6 months versus 26.4 ± 15.4 in the Sham–Stim group (*p* = 0.07). 

*Stimulation* versus *Sham Stimulation* One week after the randomization, there was a statistically significantly difference in the *Stim–Sham* group (73.3%) regarding AHI-responders compared to the *Sham–Stim* group (29.5%), a difference of 43.8% (95% CI 25.1–62.5, *p* < 0.001) between the parallel randomized groups based on intention-to-treat analysis, i.e., the two participants in the *Stim–Sham* lost to follow-up were treated as AHI non-responders (see Table 2). The effect size of the treatment difference measured by Crohn’s h was 0.99, showing a large effect size. 

For sensitivity analysis of AHI ≤ 10, the response rate was 51.1% versus 15.9% and for AHI ≤ 5, 35.6% versus 0% between the Stim–Sham and *Sham–Stim* group (see Table 3). The average ESS change from the *Stim–Sham* group was 0.4 ± 2.3 and from the *Sham–Stim* group was 5.0 ± 4.6, with a significant difference of 4.6 (95% CI of 3.1 to 6.1, *p* = 0.001) between the two groups, exceeding the two point superiority margin. The effect size of the treatment difference measured by Cohen’s d was 1.07, showing a large effect size. The EFFECT study met both co-primary endpoints.

AHI changes over time showed a significant decrease in AHI with *Stim* compared to *Sham* during the baseline, 1-week and 2-week visits (see Figure 2A). Similarly, participants reported a lower ESS with Stim as opposed to Sham during all visits (Figure 2B).

*Crossover Analysis* We assessed the change of AHI and ESS from the baseline to the 1-week and 2-week visits between the Stim–Sham and Sham–Stim groups and found no statistical evidence of a carryover effect for AHI (*p* = 0.55) or ESS (*p* = 0.23). Table 4 compares outcome measures between *Stim* and *Sham* from all complete participants under the crossover design.

Table 4 also shows other PSG parameters, highlighting the differential impact of Stim versus Sham on OSA as well as NREM versus REM sleep over the entire monitoring period. There were significant treatment differences between *Stim* and *Sham* in AHI, the apnea index, AHI in both supine and non-supine position, and AHI in both REM and non-REM (N1, N2, N3) sleep. The central and mixed apnea index did not differ between groups. The oxygen desaturation index, minimal measured oxygen saturation, and total time oxygen saturation <90% were lower with *Stim*, while mean oxygen saturation showed no difference (See Supplement for Complete PSG Data).

FOSQ improved with *Stim* compared to *Sham* (17.0 ± 3.2 versus 14.9 ± 3.6 points; *p* < 0.001). The CGI-I in the *Stim* group revealed that 76% of physician investigators rated syndromic improvement. A much stronger effect was detected in the *Sham* stimulation group, where 95% of physician investigators rated syndromic worsening (See Appendix A).

For patient and physician assessment of study arm allocation at the 1-week and 2-week visits after randomization, participants and physicians were asked to guess whether the participants were in the therapeutic stimulation group or in the sham stimulation group. Among participants, 92% guessed correctly, 3.5% guessed incorrectly, and the remaining 3.5% did not guess. Of physicians, 90% guessed correctly, 1.3% guessed wrong, and the remaining 8.7% did not guess.

The only serious adverse event in this trial was a stroke suffered by one participant in the *Sham–Stim* group during the time period of stimulation ON. He completely recovered from this event. No other adverse and severe adverse events were detected.

## 4. Discussion

The main findings of this study are that therapeutic stimulation of the hypoglossal nerve effectively treated 76.7% of participants with moderate-to-severe OSA and reduced self-reported daytime sleepiness, as compared to those with sham stimulation. After the initial sham-controlled comparison, the second crossover phase of the study showed no carryover effect. Therapeutic stimulation significantly reduced AHI, ODI and ESS and improved FOSQ. Sham stimulation led to a recurrence of OSA after one week and a return of subjective sleepiness. In addition, REM and NREM related sleep-disordered breathing were effectively treated with therapeutic stimulation versus sham stimulation. 

Several multicenter single arm prospective trials have demonstrated UAS to be highly effective [13,15,18,19,25,26]. Adequately powered, double-blinded, randomized, controlled trials are the gold standard for intervention studies eliminating the influence of unknown or immeasurable confounding variables that may otherwise lead to biased and incorrect estimates of treatment effect. Previously, only the STAR trial had included a randomized therapy withdrawal arm among therapy responders of 46 participants. The EFFECT trial provides additional support the efficacy of UAS with a randomized, sham-controlled crossover study of 89 participants. Both RCTs demonstrate consistent UAS treatment benefits in terms of OSA burden and severity, quality of life indices, and favorable sleep architectural features. 

Many patients may suffer from high cardiovascular risk if their OSA remains untreated or suboptimally treated [27]. If patients fail standard treatment with CPAP, UAS is a suitable intervention to treat sleepiness and impaired quality of life related to OSA. Because the study period was short, we did not measure any cardiovascular outcome parameters. However, it is biologically plausible that a decreased AHI and daytime sleepiness over the longer-term should be associated with improvement of cardiovascular outcomes [28]. 

Our observation that UAS effectively treats both NREM- and REM-related OSA is important. REM sleep is physiologically distinct from NREM sleep. REM-related sleep-disordered breathing is complicated by decreased lung volumes, increased upper airway collapsibility, increased sympathetic tone, and decreased respiratory drive that result in longer obstructive events, greater desaturation, and an increased rise in blood pressure at the end of an obstructive apnea [29]. Reports indicate that REM-related OSA has been independently associated with cardiovascular, neurocognitive, and metabolic risk [30]. For any treatment to be considered completely successful, OSA in REM, as well as NREM sleep, must be addressed [30].

### 4.1. Strengths of the Study

The strengths of the study include the randomized design, the use of a crossover approach, enhanced study efficiency, and increased power, allowing for the use of a smaller sample size because participants served as their own control. This trial had a low dropout rate despite the use of a crossover design that prolonged the length of the study. To the greatest possible extent, we also attempted to minimize bias by ensuring all participants and the research team were blinded to randomization assignment.

### 4.2. Limitations of the Study

This study had several limitations. The study population was predominately male (81%) and exclusively Caucasian. Our findings therefore may not be generalizable to women or the non-Caucasian population. Most participants randomized to sham stimulation became aware of the group allocation, and this may have affected subjective outcomes [31]. Because of ethical concerns and ethic committee requirements, the withdrawal period was one week. The limited withdrawal period precluded evaluating long-term consequences of subtherapeutic UAS, e.g., cardiovascular events and increased mortality associated with untreated OSA.

## 5. Conclusions

Therapeutic UAS reduced OSA severity among participants with moderate-to-severe OSA who did not tolerate CPAP. However, subtherapeutic UAS leads to the return of OSA severity within the first week of therapy withdrawal and is associated with an increase in self-reported sleepiness and a negative impact on sleep-related quality of life.

## Figures and Tables

**Figure 1 jcm-10-02880-f001:**
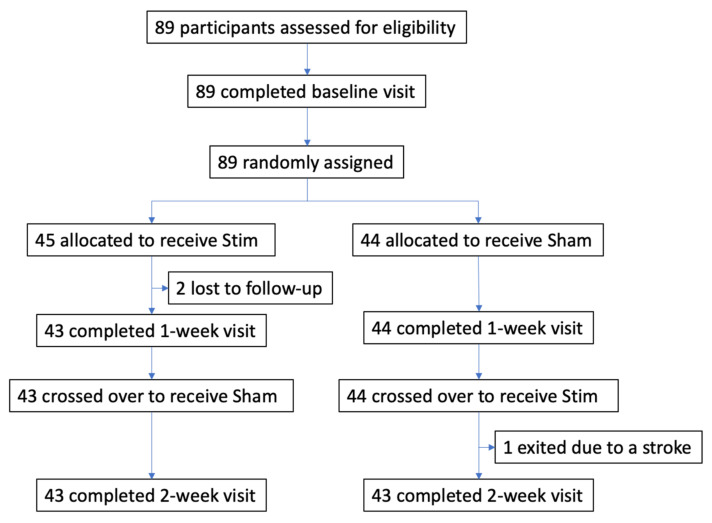
Flow diagram of the EFFECT randomized sham-controlled crossover trial.

**Figure 2 jcm-10-02880-f002:**
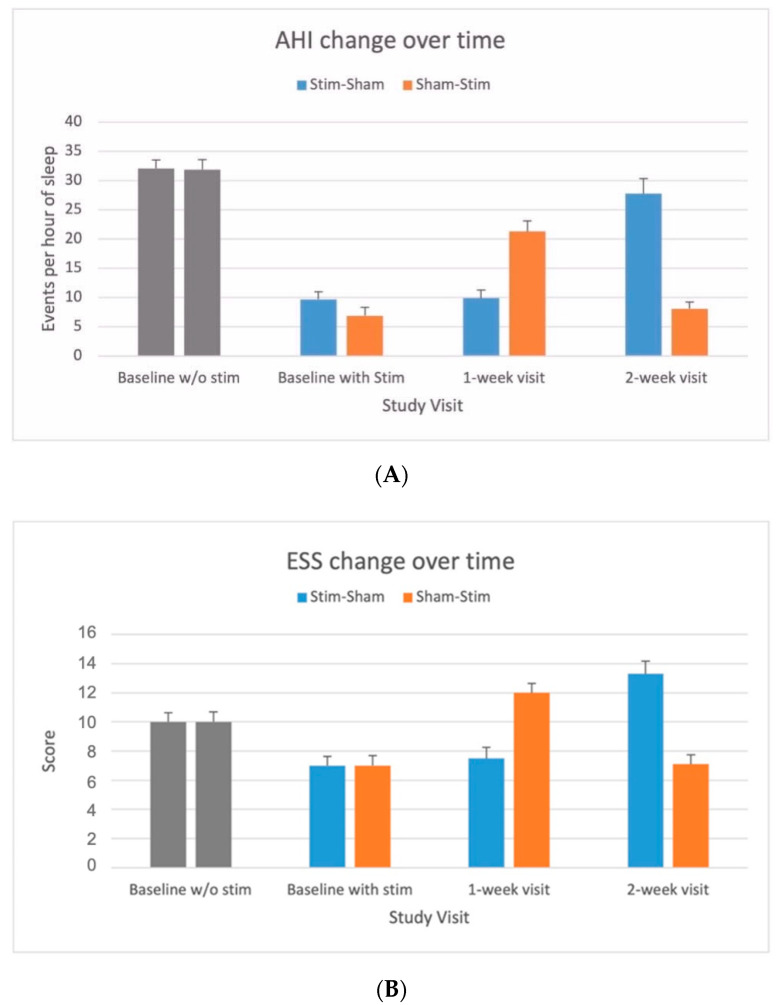
(**A**) OSA severity measured by apnea–hypopnea index (AHI) in response to upper airway stimulation versus sham. AHI values (mean and standard error bar) without stimulation before implantation and with stimulation at baseline, 1-week and 2-week visits for *stim–sham* and *sham–stim* groups. AHI values <15 events per hour of sleep considered as free of moderate-to-severe OSA [22]. (**B**). Subjective sleep propensity by Epworth Sleepiness Scale (ESS) in response to upper airway stimulation versus sham. ESS values <10 considered as free of excessive of daytime sleepiness [23].

**Table 1 jcm-10-02880-t001:** Baseline Characteristics by Randomization Group.

	All *n* = 89	Stim–Sham *n* = 45	Sham–Stim *n* = 44
Age, years	57.5 ± 9.8	58.3 ± 9.4	56.6 ± 10.4
BMI, kg/m^2^	29.2 ± 4.4	28.6 ± 3.7	29.5 ± 3.9
Male sex, %	81.0	82.2	79.5
Race, % Caucasian	100	100	100
Baseline ESS	7.0 ± 4.4	7.0 ± 4.2	7.0 ± 4.6
Baseline ESS before implantation	10.6 ± 3.8	10.0 ± 4.7	10.0 ± 4.7
Baseline AHI	8.3 ± 8.9	9.7 ± 8.5	6.9 ± 9.2
Baseline AHI before implantation	32.3 ± 11.4	32.1 ± 9.8	31.9 ± 11.4

Values are presented as mean ± standard deviation or % (*n*). AHI = apnea–hypopnea index, BMI = body mass index, ESS = Epworth Sleepiness Scale.

**Table 2 jcm-10-02880-t002:** Primary Endpoint 1: ITT comparison of proportions with AHI ≤ 15 by randomization group at visit 2.

Endpoint	Treatment 1	Treatment 2	Difference (95% CI) *p*-Value
AHI ≤ 15 (ITT)	73.3% (33/45)	29.5% (13/44)	43.8% (25.1, 62.5) < 0.001

**Table 3 jcm-10-02880-t003:** Primary Endpoint 1 Sensitivity: Comparison of proportions with AHI ≤ 10 and 5 by randomization group at visit 2.

Endpoint	Treatment 1	Treatment 2
AHI ≤ 10 (ITT)	51.1% (23/45)	15.9% (7/44)
AHI ≤ 5 (ITT)	35.6% (16/45)	0.0% (0/44)

**Table 4 jcm-10-02880-t004:** Change from baseline between Stim versus Sham in all participants with moderate to severe OSA.

Parameter	Stim (*n* = 86)	Sham (*n* = 86)	Treatment Difference	*p*-Value
PSG Parameters				
AHI (events/h)	0.6 (−1.8, 2.9)	16.1 (13.7, 18.4)	−15.5 (−18.3, −12.8)	<0.001
ODI (events/h)	0.6 (−1.9, 3.0)	12.7 (10.3, 15.2)	−12.2 (−14.8, −9.6)	<0.001
Apnea index (events/h)	0.5 (−1.2, 2.3)	8.9 (7.2, 10.7)	−8.4 (−10.6, −6.2)	<0.001
AHI in supine position (events/h)	2.2 (−2.3, 6.6)	23.8 (19.4, 28.2)	−21.6 (−27.2, −16.0)	<0.001
AHI in non-supine position (events/h)	−0.1 (−3.2, 2.9)	3.1 (0.1, 6.1)	−3.3 (−6.4, −0.1)	0.044
AHI in REM sleep (events/h)	2.0 (−1.6, 5.6)	17.1 (13.5, 20.6)	−15.1 (−19.7, −10.5)	<0.001
AHI in non-REM sleep (events/h)	0.0 (−2.4, 2.5)	15.7 (13.3, 18.2)	−15.7 (−18.5, −12.8)	<0.001
Central Apnea Index (events/h)	0.1 (−0.1, 0.4)	0.3 (0.0, 0.5)	−0.1 (−0.4, 0.1)	0.285
Mixed Apnea Index (events/h)	0.1 (−0.3, 0.4)	0.3 (−0.1, 0.6)	−0.2 (−0.6, 0.2)	0.355
Central Mixed Apnea Index (events/h)	−0.0 (−0.8, 0.7)	0.4 (−0.3, 1.1)	−0.4 (−1.2, 0.4)	0.283
Hypopnea Index (events/h)	0.0 (−1.6, 1.6)	7.0 (5.4, 8.6)	−7.0 (−8.9, −5.1)	<0.001
Minimal measured SaO2 (%)	−0.9 (−1.9, 0.2)	−4.0 (−5.0, −3.0)	3.1 (2.1, 4.2)	<0.001
Mean SaO2 (%)	−0.2 (−0.9, 0.4)	−0.5 (−1.2, 0.1)	0.3 (−0.5, 1.1)	0.493
Total time SaO2 <90%	2.4 (−1.7, 6.4)	9.0 (4.9, 13.0)	−6.6 (−11.2, −2.0)	0.005
Quality of life measures				
ESS (points)	0.2 (−0.7, 1.1)	3.5 (2.6, 4.4)	−3.3 (−4.4, −2.2)	<0.001
FOSQ (points)	0.2 (−0.5, 0.9)	−1.9 (−2.6, −1.2)	2.1 (1.4, 2.8)	<0.001

## Data Availability

All the individual patient data collected during the trial will be shared. In addition, the study protocol and statistical analysis plan will be available as well. The data will be made available within 12 months of publication. All available data can be obtained by contacting the corresponding author (clemens.heiser@tum.de). It will be necessary to provide a detailed protocol for the proposed study, to provide the approval of an ethics committee, to supply information about the funding and resources one has to carry out the study, and to consider inviting the original authors to participate in the re-analysis.

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
