# Peer review of "Effect of Upper Airway Stimulation in Patients with Obstructive Sleep Apnea (EFFECT): A Randomized Controlled Crossover Trial"

_jcm, 2021, doi:10.3390/jcm10132880_

Round 1

Reviewer 1 Report

This is a very well planned, conducted and executed randomized controlled crossover trial. Topic is extremely important and up-to-date, especially for a growing population of patients, who do not tolerate CPAP. However, some minor criticism should be addressed:

L46 - there are a few newly published articles worth mentioning briefly within this part of Introduction section, as authors skipped OSA screening (MMP, CBCT, questionnaires) prior to PSG completely, please incoroporate and cite https://www.mdpi.com/2076-3417/11/9/3764 and https://www.tandfonline.com/doi/full/10.1080/08869634.2020.1765602

L195 -'AHI values < 15 events per hour of sleep considered as free of moderate-to-severe OSA' - please provide a relevant reference

L200 - 'ESS values < 10 considered as free of excessive of daytime sleepiness' - please provide a relevant reference

L216 - Table 2 should be presented as at least two or more different graphs/figures - PSG and QoL values should be taken separately; it is slightly messy, not very clear and difficult to read now. Consider linear graphs or bar-charts, heatmap will do nicely, either.

L269 and L275 - strengths and limitations of the study should get separate subheadings for clarity

L285 - 'did not adhere' is vague, consider replacement

L290 - www.mdpi.com/xxx/s1 link is invalid, please provide a relevant one

L315 - furthermore we ?

Author Response

This is a very well planned, conducted and executed randomized controlled crossover trial. Topic is extremely important and up-to-date, especially for a growing population of patients, who do not tolerate CPAP. However, some minor criticism should be addressed:

L46 - there are a few newly published articles worth mentioning briefly within this part of Introduction section, as authors skipped OSA screening (MMP, CBCT, questionnaires) prior to PSG completely, please incorporate and cite https://www.mdpi.com/2076-3417/11/9/3764 and https://www.tandfonline.com/doi/full/10.1080/08869634.2020.1765602

Response: We would like to thank the reviewer for suggesting the additional OSA screening prior to PSG for upper airway stimulation. For patients receiving upper airway stimulation, the current patient screening includes intolerance to CPAP, moderate to severe OSA and upper airway anatomical exam with drug induced sleep endoscopy. We refer the reviewer to the reference [10], which includes clinical evidence that support these patient selection criteria for upper airway stimulation. In fact, patients with PAP-intolerance were already diagnosed with OSA before starting PAP-therapy. Therefore, OSA screening would not make sense in such cases.

L195 -'AHI values < 15 events per hour of sleep considered as free of moderate-to-severe OSA' - please provide a relevant reference

Response: We added the AASM Manual Scoring as reference [22]

L200 - 'ESS values < 10 considered as free of excessive of daytime sleepiness' - please provide a relevant reference

Response: We added the report Weaver et al as reference [23]

L216 - Table 2 should be presented as at least two or more different graphs/figures - PSG and QoL values should be taken separately; it is slightly messy, not very clear and difficult to read now. Consider linear graphs or bar-charts, heatmap will do nicely, either. 

Response: The co-primary study outcome measures of AHI and ESS are in bar-charts in Figure 2A and 2B, as the reviewer recommended. We have included main PSG parameters and Quality of life measures in Table 2 for a comprehensive summary of the study outcome. Separately, we have summarized sleep architecture results in the supplement (Table S1). We will continue to work with the journal editorial staff during final editing to improve clarity of these tables.

L269 and L275 - strengths and limitations of the study should get separate subheadings for clarity

Response: Thanks for the comment. We added the subheadings.

L285 - 'did not adhere' is vague, consider replacement

Response: We totally agree with the reviewer and changed adhere to tolerate.

L290 - www.mdpi.com/xxx/s1 link is invalid, please provide a relevant one

Response: We will work with the journal editorial staff to update the link when it is assigned.

L315 - furthermore we ?

Response: We would like to thank the reviewer for the note. We added the missing sentence to the acknowledgments. 

Reviewer 2 Report

The authors have conducted a very elegant RCT with HNS. This type of highly well-designed trial is absolutely necessary in order to change the reports like the EUnetHTA that was so harmful to the inclusion of the HNS in other countries. Most of the reasons for this bad report was the lack of these well-designed trials to prove the effect of HNS.

After reading this manuscript I have some comments

Line 128-130 you say: If the primary efficacy endpoints were met, the endpoints were then analyzed according to the crossover design. So, what happens if the primary outcomes were not met? How did you analyze that? 

Line 177: “After One Week One week after the randomization”, I believe that One Week should be only written once, and if it’s ok, I don’t understand the sentence so try to change it so everyone can.

Your results are really good but the goal was: we designed a randomized control trial to prospectively enroll UAS recipients regardless of their therapy response and I don’t see any non-responders to HNS, I don’t know, maybe they’re very few amongst the responders that the mean AHI is really good, so maybe you should show them to those technicians that are going to do the new reviews with your data. 

Please explain if you did a per-protocol analysis. See the comments the technicians did when I was discussing the previous trial: “In the study by Woodson how can intention to treat approach can be done? There’s no change in the group, so this cannot apply, why should something that cannot apply help to rate less? Ok that randomization is not explained, rate less, ok with blinding, although I can assure you that if you don’t turn it on, or even if you turn it on but there’s no stimulation, with such a big change in sleepiness and all that the patients are going to know so the blinding is not effective, but intention to treat does not apply here as there’s no deviation from the initial plan, results are the same. This cannot make this study worse.
You’re the expert in rating studies, so tell us which kind of study we should perform so you think that the evidence is high and HGNS is useful. I’m sure that everybody that is implanting and watching how this therapy is helping to the patients will like to do it so this technology is spread and accepted without questions”  

And their answer:

“Intention-to-treat analysis is a comparison of the treatment groups that includes all patients as originally allocated after randomization. Per- protocol analysis is a comparison of treatment groups that includes only those patients who completed the treatment originally allocated. If done alone, this analysis leads to bias. The selected RCT by Woodson el al., presents a selection bias as only responders to UAS system from the STAR cohort were randomized. Likewise, the authors did not report a specific statement on the analysis approach for the missing observations which is required to evaluate the quality of RCT”... depressing

So please write this statement, so they cannot downrate the quality of the study. 

Author Response

The authors have conducted a very elegant RCT with HNS. This type of highly well-designed trial is absolutely necessary in order to change the reports like the EUnetHTA that was so harmful to the inclusion of the HNS in other countries. Most of the reasons for this bad report was the lack of these well-designed trials to prove the effect of HNS.

After reading this manuscript I have some comments

Line 128-130 you say: If the primary efficacy endpoints were met, the endpoints were then analyzed according to the crossover design. So, what happens if the primary outcomes were not met? How did you analyze that? 

Response: We would really like to thank the reviewer for this valuable comment. When the primary endpoints were met, the type 1 error is preserved, and the crossover study can be analyzed. For this study, both co-primary endpoints were met. If, however, the primary endpoints were not met, then there would be no remaining type 1 error to analyze the crossover outcomes, and the outcome could only be presented in descriptive statistics.

Line 177: “After One Week One week after the randomization”, I believe that One Week should be only written once, and if it’s ok, I don’t understand the sentence so try to change it so everyone can.

Response: Thanks for this valuable comment and we would like to thank reviewer for the note. We delete the double “one week”, which makes it now easier to understand.

Your results are really good but the goal was: we designed a randomized control trial to prospectively enroll UAS recipients regardless of their therapy response and I don’t see any non-responders to HNS, I don’t know, maybe they’re very few amongst the responders that the mean AHI is really good, so maybe you should show them to those technicians that are going to do the new reviews with your data. 

Response: We recruited all implant recipients in a consecutive manner to participate independently if they have responded to therapy or not. In Line 96 of the “Materials and Methods”, we have highlighted “consecutively” with italics in responding to the reviewer’s comment.

Please explain if you did a per-protocol analysis. See the comments the technicians did when I was discussing the previous trial: “In the study by Woodson how can intention to treat approach can be done? There’s no change in the group, so this cannot apply, why should something that cannot apply help to rate less? Ok that randomization is not explained, rate less, ok with blinding, although I can assure you that if you don’t turn it on, or even if you turn it on but there’s no stimulation, with such a big change in sleepiness and all that the patients are going to know so the blinding is not effective, but intention to treat does not apply here as there’s no deviation from the initial plan, results are the same. This cannot make this study worse.
You’re the expert in rating studies, so tell us which kind of study we should perform so you think that the evidence is high and HGNS is useful. I’m sure that everybody that is implanting and watching how this therapy is helping to the patients will like to do it so this technology is spread and accepted without questions”  

And their answer:

“Intention-to-treat analysis is a comparison of the treatment groups that includes all patients as originally allocated after randomization. Per- protocol analysis is a comparison of treatment groups that includes only those patients who completed the treatment originally allocated. If done alone, this analysis leads to bias. The selected RCT by Woodson el al., presents a selection bias as only responders to UAS system from the STAR cohort were randomized. Likewise, the authors did not report a specific statement on the analysis approach for the missing observations which is required to evaluate the quality of RCT”... depressing

So please write this statement, so they cannot downrate the quality of the study. 

Response: The primary endpoints were met based on both intention-to-treat and per-protocol analyses. We have included the intention-to-treat analysis in the result section as the reviewer recommended.

Primary Endpoint 1: ITT Comparison of proportions with AHI <= 15 by randomization group at Visit 2

Endpoint

Treatment 1

Treatment 2

Difference (95% CI) p-value

AHI <= 15 (ITT)

73.3% (33/45)

29.5% (13/44)

43.8% (25.1, 62.5) <.001

Reviewer 3 Report

Author and colleagues have designed RCT in order to overcome the severe limitation of STAR trial in order to enroll UAS recipients regardless of their response. Despite this strength, there are some concerns that you have to handle, which, I believe, will improve the readability of this study.

Major

  1. Many readers are interested in the response rate when the responder is defined not only as AHI<15/h, but also as AHI<10/h or AHI<5/h. It should be easy to show these data at the authors' institution.
  2. Result (Table2) Why don't you present data on arousal index and sleep architecture? Many readers are interested not only in changes in ahi, but also in the effects of UAS on arousal threshold and sleep architecture. It should be easy to show these data at the authors' institution.
  3. Discussion (paragraph 3), Conclusions(Line 4) In recent years, REM-related OSA and NREM-related OSA have been recognized as clinically important OSA phenotypes. In addition, the diagnostic criteria for these clinical phenotypes have been largely established. Therefore, it is highly problematic to assume that UAS affects these clinical phenotypes based solely on changes in AHI during REM and non-REM sleep. Please review your data based on previously reported definitions of REM-related OSA and NREM-related OSA. It should be easy to show these data at the authors' institution.

Author Response

Author and colleagues have designed RCT in order to overcome the severe limitation of STAR trial in order to enroll UAS recipients regardless of their response. Despite this strength, there are some concerns that you have to handle, which, I believe, will improve the readability of this study.

Major

  1. Many readers are interested in the response rate when the responder is defined not only as AHI<15/h, but also as AHI<10/h or AHI<5/h. It should be easy to show these data at the authors' institution.

Response: We have added response rate using AHI < 10 or 5/h in the result section as the reviewer recommended.

Primary Endpoint 1 Sensitivity: Comparison of proportions with AHI <= 10 and 5 by randomization group at Visit 2

Endpoint

Treatment 1

Treatment 2

AHI <= 10 (ITT)

51.1% (23/45)

15.9% (7/44)

AHI <= 5 (ITT)

35.6% (16/45)

0.0% (0/44)

  1. Result (Table2) Why don't you present data on arousal index and sleep architecture? Many readers are interested not only in changes in ahi, but also in the effects of UAS on arousal threshold and sleep architecture. It should be easy to show these data at the authors' institution.,

Response: We would like to thank the reviewer for this valuable comment. We already provided this data in the supplemental material.

  1. Discussion (paragraph 3), Conclusions(Line 4) In recent years, REM-related OSA and NREM-related OSA have been recognized as clinically important OSA phenotypes. In addition, the diagnostic criteria for these clinical phenotypes have been largely established. Therefore, it is highly problematic to assume that UAS affects these clinical phenotypes based solely on changes in AHI during REM and non-REM sleep. Please review your data based on previously reported definitions of REM-related OSA and NREM-related OSA. It should be easy to show these data at the authors' institution.

Response: REM-related OSA is an important predictor of cardiovascular risk. We agree with the reviewer that comprehensive phenotype analysis is warranted besides AHI. As the current study focuses on AHI and ESS as the primary outcome measures, we need to review source sleep study reports to analyze REM-related parameters, and plan to include it in a separate report. As a result, we have removed the REM-related sleep disordered breathing statement from the conclusion of the current report.    

Round 2

Reviewer 3 Report

The content of the manuscript has been greatly improved, but there are still several areas that require minor revision. 1.Table S1 N1, N2, N3, REM, NREM, wake should be expressed as a percentage of TST, so that the comparison between stim and sham will be clearer. 2.Table S1 Is sleep efficiency being calculated correctly? Sleep efficiency in Stim and Sham are too low. As we know, Sleep efficiency is the percentage of time spent asleep while in bed.

Author Response

We would like to thank the reviewer for the valuable comments and kindly response to them. We highly appreciate the feedback:

  • With table 1 the differences should be expressed and its CI to make the table congruent. For us it seems to be more clearly to the reader.
  • Yes, sleep efficiency is calculated correctly. Again it shows the differences and not the numbers... 

Hopefully this helped to clarify the comments and we are looking forward to your response.

Kind regards